# Fast lithium growth and short circuit induced by localized-temperature hotspots in lithium batteries

Yangying Zhu [1,6], Jin Xie [1,2,6], Allen Pei [1], Bofei Liu[1], Yecun Wu [1,3], Dingchang Lin [1], Jun Li[1,4], Hansen Wang [1], Hao Chen[1], Jinwei Xu[1], Ankun Yang [1], Chun-Lan Wu [1], Hongxia Wang [1], Wei Chen[1] & Yi Cui [1,5]

Fast-charging and high-energy-density batteries pose significant safety concerns due to high rates of heat generation. Understanding how localized high temperatures affect the battery is critical but remains challenging, mainly due to the difficulty of probing battery internal temperature with high spatial resolution. Here we introduce a method to induce and sense localized high temperature inside a lithium battery using micro-Raman spectroscopy. We discover that temperature hotspots can induce significant lithium metal growth as compared to the surrounding lower temperature area due to the locally enhanced surface exchange current density. More importantly, localized high temperature can be one of the factors to cause battery internal shorting, which further elevates the temperature and increases the risk of thermal runaway. This work provides important insights on the effects of heterogeneous temperatures within batteries and aids the development of safer batteries, thermal management schemes, and diagnostic tools.

[1] Department of Materials Science and Engineering, Stanford University, Stanford, CA 94305, USA. [2] School of Physical Science and Technology, ShanghaiTech University, 201210 Shanghai, China. [3] Department of Electrical Engineering, Stanford University, Stanford, CA 94305, USA. [4] Department of Chemistry, Stanford University, Stanford, CA 94305, USA. [5] SLAC National Accelerator Laboratory, Stanford Institute for Materials and Energy Sciences, 2575 Sand Hill Road, Menlo Park, CA 94025, USA. [6] These authors contributed equally: Yangying Zhu, Jin Xie. Correspondence and requests for materials should be addressed to Y.C. (email: yicui@stanford.edu)

Energy storage with batteries is critical to enable renewable energy technologies and environmental sustainability[1,2]. Significant progress has been made on the development of rechargeable lithium-based batteries in recent decades[3–6]. However, the increasing charging rate and energy density pose significant safety concerns as self-heating becomes a non-negligible effect[7,8]. While the role of uniform temperature on lithium growth morphology[9], cyclability[10], and aging rate[11] has been studied previously, batteries in realistic situations generally operate with non-uniform temperature and sometimes can have localized-temperature hotspots from internal or external heat sources[12,13], or from manufacturing nonuniformity and defects[14]. How localized high temperature affects battery operations is not yet understood.

Among the challenges to study local-temperature effects is the difficulty of probing the internal temperature of batteries with high spatial resolution. Temperature measurement techniques employed in batteries are typically remote (e.g., sensors attached to battery external packaging[15,16]) or macroscopic (e.g., thermocouples and infrared imaging[16–18]). However, the small length scales of battery electrode materials and their electrochemical processes require temperature sensing at a more microscopic level. In particular, in the event of thermal runaway, which can cause catastrophic fire or explosion and is typically caused by battery internal shorting[18,19], capturing the local-temperature response provides valuable information to aid fundamental understanding of failure mechanisms and the development of thermal management strategies.

Lithium (Li) metal has recently been studied intensively as an attractive anode with highest possible specific capacity and understanding all the factors affecting its growth is critical for Li metal batteries and the safety of the existing lithium-ion batteries[20–22]. In this work, we investigated the effect of local-temperature hotspots on Li metal growth and accordingly proposed a temperature-induced battery shorting mechanism as one possible concern when considering battery safety. Localized high temperature was created internally in a Li battery with a laser and measured using a micro-Raman spectroscopy platform. Li deposition rate was found to be orders of magnitude faster on the hotspot due to the enhanced surface exchange current density. Based on this observation, we further demonstrated that localized high temperature can be one of the factors to cause battery shorting, which was supported by optical visualization and simultaneous voltage-current and local temperature response measurements. The temperature measurement platform opens new doors for detailed thermal characterization of energy storage devices.

## Results

**Raman spectroscopy for hotspot temperature measurement**. To investigate the effect of an internal hotspot on lithium growth behavior, we probed Li batteries using Raman spectroscopy, which provides simultaneous local heating and temperature sensing capabilities with high spatial resolution. As a proof of concept, we first investigated a coin cell modified with a thin glass window (1 cm$^2$ disk with a thickness of 145 μm) transparent to visible laser light (see Methods). As illustrated in Fig. 1a, the battery consists of a 170 nm thermally evaporated copper (Cu) layer as the working electrode, a 50 μm Li foil as the counter electrode, and carbonate based additive-free electrolyte. Prior to Cu deposition, graphene, which has a temperature-dependent Raman shift[23,24], was transferred onto the glass as a temperature indicator (see Methods). Among various temperature-sensing nanomaterials such as silicon nanostructures[25] or carbon nanotubes[26], graphene was selected because its ultra-thin thickness ensures excellent thermal contact with Cu, negligible thermal

mass, and a flat Cu-electrolyte interface. The thickness of the Cu (170 nm) was chosen to offer high enough electrical conductivity while ensuring that the temperature on the Cu-graphene interface was similar to the Cu-electrolyte interface (see Supplementary Table 1). Laser (wavelength of 532 nm) from a ×100 objective was focused on the Cu-graphene interface. Absorption of the laser energy created a hotspot on the current collector, and the temperature of the hotspot was determined by the temperature-dependent Raman shift of the graphene.

More specifically, the first step to realizing temperature measurement was to calibrate the temperature dependence of the G-band Raman shift of graphene sandwiched between glass and Cu. As illustrated in Fig. 1b (inset), the glass-graphene-Cu trilayer was allowed to reach thermal equilibrium with a constant temperature stage (estimated accuracy of ±0.5 °C). The graphene was then excited with a visible laser ($\lambda = 532$ nm) at a low excitation power of 0.3 mW to avoid local heating of the sample above the steady-state temperature setpoint. High spectral resolution was achieved with an 1800 gr/mm grating, a long focal length spectrometer, and a high-resolution CCD camera (HORIBA Scientific LabRAM HR Evolution spectrometer). Each spectrum was collected for 120 s and the G-band peak position was fitted with the Voigt profile[27]. Figure 1b shows that the Raman peak position changed linearly over a temperature range of 30–110 °C. The uncertainty of the peak position from multiple measurements was ~0.06 cm$^{-1}$. The linear temperature coefficient $A$ (defined as $A = \Delta\omega/\Delta T$ where $\omega$ is the Raman peak position and $T$ is the temperature) was calculated to be $-0.0559 \pm 0.009$ cm$^{-1}$ °C$^{-1}$ from linear fit (dotted line in Fig. 1b), which is similar to literature values of graphene on copper[28].

After the temperature coefficient was calibrated, the glass-graphene-Cu trilayer was assembled into coin cells as shown in Fig. 1a. We locally heated the Cu with a 532 nm laser and measured the G-band peak positions $\omega_i$ at various laser powers of $P_i$, while the bulk cell remained at room temperature. The hotspot temperature $T_i$ was calculated from equation $\omega_i - \omega_0 = A(T_i - T_0)$ where $A$ is the calibrated temperature coefficient ($-0.0559 \pm 0.009$ cm$^{-1}$ °C$^{-1}$) and $\omega_0$ is the Raman peak position at the reference room temperature $T_0$. Figure 1c shows that as the laser power was increased from 0 to 20.1 mW, the hotspot temperature rose linearly from room temperature to ~119 °C. The uncertainty in temperature prediction (Fig. 1c) accounts for multiple measurement error, uncertainty in the temperature coefficient $A$ from calibration, and the accuracy of the temperature stage temperature. It should be noted that the Raman shift can be affected by strain as well, and that the differences between the strain in the uniformly-heated calibration and the locally heated experiment may cause artifacts in the temperature measurement. However, this effect is estimated to be small due to the small thermal expansion coefficient of the glass substrate. To further validate the measured hotspot temperature in Fig. 1c, a thermal model accounting for spreading of the heat generated by the laser to the surrounding Cu, glass, and electrolyte was developed in COMSOL Multiphysics (see Supplementary Notes), and the simulated hotspot peak temperature (solid line in Fig. 1c) shows good agreement with the Raman measurement (dots in Fig. 1c).

**Lithium growth on the hotspot**. While Li deposition morphology at uniform temperature has been investigated previously[9–11], the effect of local-temperature variation has been less studied. To understand how local hotspots affect the battery, Li growth behavior in the presence of a hotspot with controlled temperature was investigated on the Raman spectroscopy platform and examined by scanning electron microscopy (SEM). In the experiment, a constant amount of charge was applied at the same

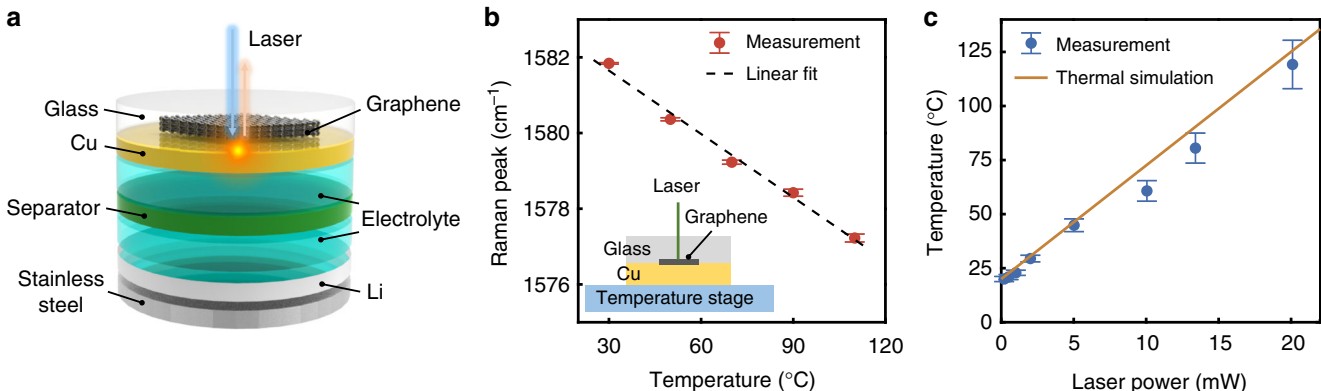

**Fig. 1** Experimental setup. **a** Schematic (not to scale) of a modified coin cell with an optically transparent glass window for laser access to graphene as a temperature indicator and the thermally evaporated Cu current collector. **b** G-band Raman peak position of the graphene as a function of the temperature. The temperature coefficient was obtained from the slope of the linear fit (dashed line). The inset shows the schematic of the calibration setup. **c** Temperature of the hotspot on Cu generated by a 532 nm laser in a coin cell as a function of the laser power. The dots are experimental results and the solid line is from thermal simulation in COMSOL Multiphysics

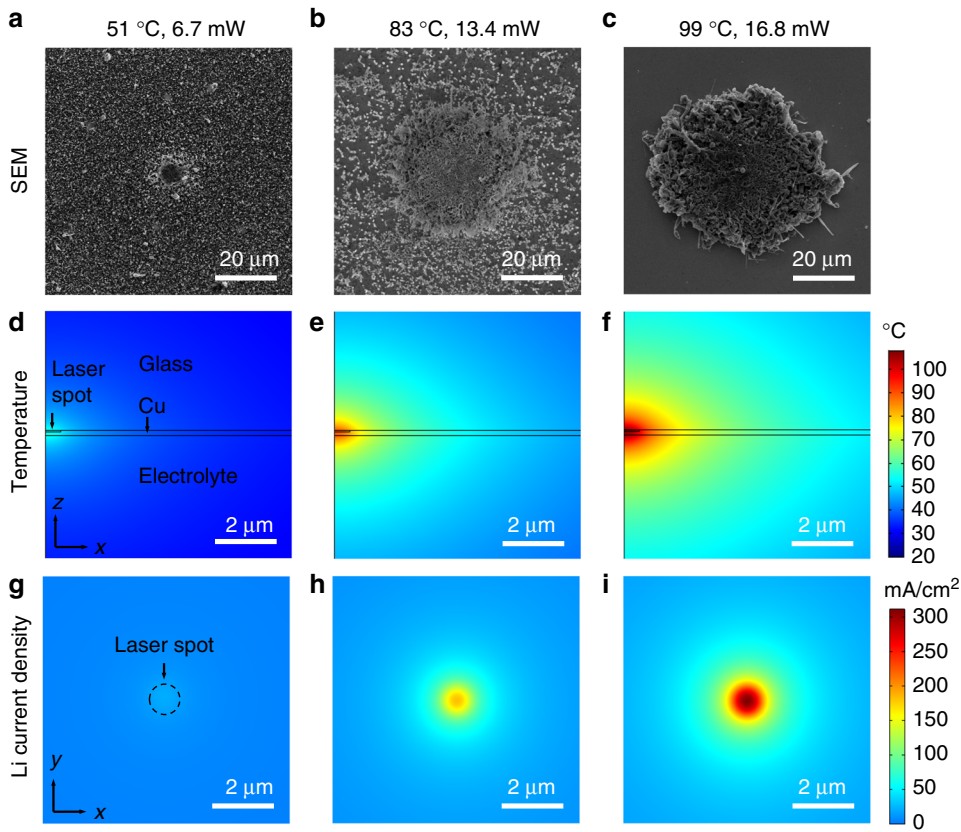

**Fig. 2** Lithium deposition on hotspots. SEM images (top-down view) of Li deposited on Cu with hotspot temperatures of **a** 51 °C at a laser power of 6.7 mW, **b** 83 °C at 13.4 mW, and **c** 99 °C at 16.8 mW, respectively. The corresponding (cross-sectional view) temperature distribution from simulation near the laser spot with powers of **d** 6.7 mW, **e** 13.4 mW, and **f** 16.8 mW. The simulated Li deposition rate on the Cu surface (top-down view) with laser heated hotspot temperatures of **g** 51 °C at 6.7 mW, **h** 83 °C at 13.4 mW, and **i** 99 °C at 16.8 mW, respectively

Li-plating rate of $1\,\mathrm{mA\,cm^{-2}}$ for 2 min for coin cells with different laser heating power. The spot size of the focused laser beam was ~500 nm in radius. After Li plating, batteries were immediately disassembled inside the glove box and the morphology of Li deposited on the hotspot and the surroundings was characterized in SEM (see details in Methods). As shown in Fig. 2a–c, for laser powers of 6.7, 13.4, and 16.8 mW, which correspond to hotspot temperatures of 51, 83, and 99 °C, respectively, according to the Raman measurement, Li deposited significantly faster on the hot

region (center of the SEM images). As the hotspot temperature increased, more Li was grown on the hotspot with respect to the surrounding lower-temperature background. Localized Li growth was also observed on a thin-film-metal line heater (see Supplementary Fig. 1), but was not present in coin cells with uniform temperatures (see Supplementary Fig. 2).

To understand the observed non-uniform Li deposition, we simulated the initial temperature distribution and Li deposition current density distribution in COMSOL Multiphysics. The inputs

of the temperature model only include thermophysical properties and geometries of the cell materials with no fitting parameters (see Supplementary notes, Supplementary Table 2, and Supplementary Figs. 3–7). Figure 2d–f shows the temperature distribution of the cross section of the axisymmetric half-cell including a laser spot (500 nm in radius) and the heat spreading media for incident laser powers of 6.7, 13.4, and 16.8 mW (absorption of 0.4 for 532 nm laser on Cu[29]). The peak temperature of the hotspots from simulation increases with the laser power from 55 °C (Fig. 2d), 90 °C (Fig. 2e) to 108 °C (Fig. 2f), which agrees well with the measured temperatures. The low thermal conductivity (on the order of 1 W m$^{-1}$ K$^{-1}$) of the glass and electrolyte, and the thin (170 nm) Cu film (350 W m$^{-1}$ K$^{-1}$) contribute to the high peak temperature (see Supplementary Table 3 and Supplementary Fig. 8). The temperature rise is localized, primarily as a result of the small heat source. In particular, temperature decays to half of the peak value (half width at half maximum, HWHM) at the same distance of $r = 3.7$ μm for all laser powers. The temperature profiles normalized with respect to the peak values are identical, due to linearity of the heat equation[30]. The majority of the cell tens of microns away from the hotspot remains unheated at room temperature. The temperature distribution can also be affected by the spot size of the heat source (see Supplementary Fig. 9); however, in the experiment the focused spot size of the laser was not adjustable. In addition, the simulated temperature is only for the equilibrium state before Li deposition occurs. The deposited Li can contribute to heat spreading and lowers the peak temperature (see Supplementary Fig. 10).

The temperature distribution was imported to the electrochemical model to simulate the effect of the hotspot on Li deposition. A cell with a Cu working electrode and Li counter electrode of areas 1 cm$^2$ was constructed, and a 1 mA cm$^{-2}$ current density was applied to the Li counter electrode to observe variations in local deposition rate at the Cu electrode (see Supplementary Note and Supplementary Fig. 11). The exchange current density of Li/Li$^+$ redox at the electrode-electrolyte interface was scaled as a function of the Cu surface temperature through adjusting the rate constant, $k$, using an Arrhenius relation[31,32],

$$k = A e^{\frac{-E_a}{RT(r)}} \qquad (1)$$

where $R$ is the gas constant and $T(r)$ is the temperature in Kelvins of the Cu surface at a distance $r$ away from the center of the laser spot. $E_a$ is a generalized activation energy for a lithium redox event, which was experimentally determined to be 73.5 kJ mol$^{-1}$ through measurements by fast scans using ultramicroelectrodes (see Supplementary Fig. 12). $A$ is a scaling factor, which was fit through the same experimental results, giving an overall expression for the exchange current density as a function of temperature:

$$j_0 = e^{\left(\frac{-E_a}{RT(r)}+32.01\right)} \qquad (2)$$

Here, we have only considered the effect of temperature on the electrochemical kinetics and neglected effects on electrolyte conductivity due to the significant dominance of the exponentially increasing rate constant with temperature over effects from enhanced ion mobility in the electrolyte (see Supplementary Note and Supplementary Figs. 13–14).

The local Li deposition rate drastically increases with increased laser power and corresponding local electrode heating (Fig. 2g–i). Peak current densities at the center of the laser spot are 21.5, 182.3, and 311.2 mA cm$^{-2}$ for laser powers of 6.7 mW (51 °C), 13.4 mW (83 °C), and 16.8 mW (99 °C), respectively. These values are 1–2 orders of magnitude higher than the background current density (1 mA cm$^{-2}$). The average current densities (and thus also Li deposition capacities) within a 10 μm radius around the hotspot are also significantly higher (5.8, 27.2, and 44.9 mA cm$^{-2}$ for

hotspot temperatures of 51, 83, and 99 °C, respectively) than the background. The current density drops to half its maximum value at 1 μm from the center for the 6.7 mW laser as compared to 0.68 μm for the 16.8 mW laser (see Supplementary Figs. 15–16). Thus, in addition to increasing the peak local current density, increasing the local maximum temperature also decreases the width of the enhancement in current density, effectively concentrating the lithium-ion flux close to the hotspot. This phenomenon can provide one explanation for the seemingly larger locally enhanced Li deposit seen in Fig. 2c for the highest laser power; the combination of the increased deposition current/capacity and the more localized enhancement could force the lithium to grow and be flattened against the separator and counter electrode, a common occurrence seen in coin cells with high areal capacities of deposited Li. Furthermore, the locally increased exchange current densities could cause the already favored nucleation and deposition of Li on existing Li over Cu to be even more selective, causing the overgrowth of Li observed and surrounding bare Cu in Fig. 2c. Another factor that could cause the deposited Li to be much larger than the simulated initial thermal and electrochemical fields is that the deposited Li could spread the heat, which effectively widens the temperature peak for subsequent Li deposition (Supplementary Fig. 10). In all cases, it is clear that a local heating event can drastically enhance localized deposition of Li. The exponential nature of the increases in reaction kinetics with temperature highlight the sensitivity of the electrochemistry within lithium-ion or lithium metal batteries to temperature fluctuations.

**Hotspot-induced battery shorting and local-temperature sensing.** While it has been well understood that internal shorting can generate hotspots and cause thermal runaway[7], the fast Li growth on the hotspots lead us to propose that internal local high temperature could in reverse be a mechanism to trigger battery shorting. We first verify this hypothesis by simultaneous optical visualization and voltage-current measurement of an optical battery cell. With the ability to thermally seed and grow Li at any specific location through laser heating, we further detected the local-temperature rise of a hotspot-triggered internal shorting event with a fabricated micro temperature detector. This local-temperature detection would otherwise be challenging to achieve without being able to induce a short at will and knowing where the short occurred.

Simultaneous visualization and voltage-current measurement was carried out on an optical cell (Fig. 3a), which consists of a Cu foil (thickness of 12 μm and width of 3 mm) as the working electrode and lithium cobalt oxide (LCO) on an Al foil as the counter electrode (see Methods). A hotspot (~43 °C, see Supplementary Fig. 17) was generated near the edge of the Cu with a laser (13.4 mW through a ×10 magnification objective to include a large field of view), and an optical image was captured through the same ×10 objective every 40 s in alternation with the laser light source. We first examined the battery under galvanostatic charging at 30 μA (Fig. 3b). On the hotspot (Fig. 3c), a Li chunk quickly formed (Fig. 3d, see Supplementary Movie 1), whereas Li growth rates in the surrounding remained slow. At 1480 s, Li touched the counter electrode (Fig. 3f). Simultaneously, the cell voltage dropped (Fig. 3b, onset of shorting) and began to fluctuate as charging continued (Fig. 3g). The in situ visualization proved our hypothesis that local high temperature can lead to battery shorting.

Potentiostatic charging represents another realistic charging mode in battery operation, similar if not more important than the galvanostatic charging mode. For this mode, we detected local-temperature response at the shorting location by embedding a thin-film resistance temperature detector (RTD) in the gap between two electrodes (Fig. 4a, see Method). To ensure that shorting occurs precisely on top of the RTD, we aligned the laser

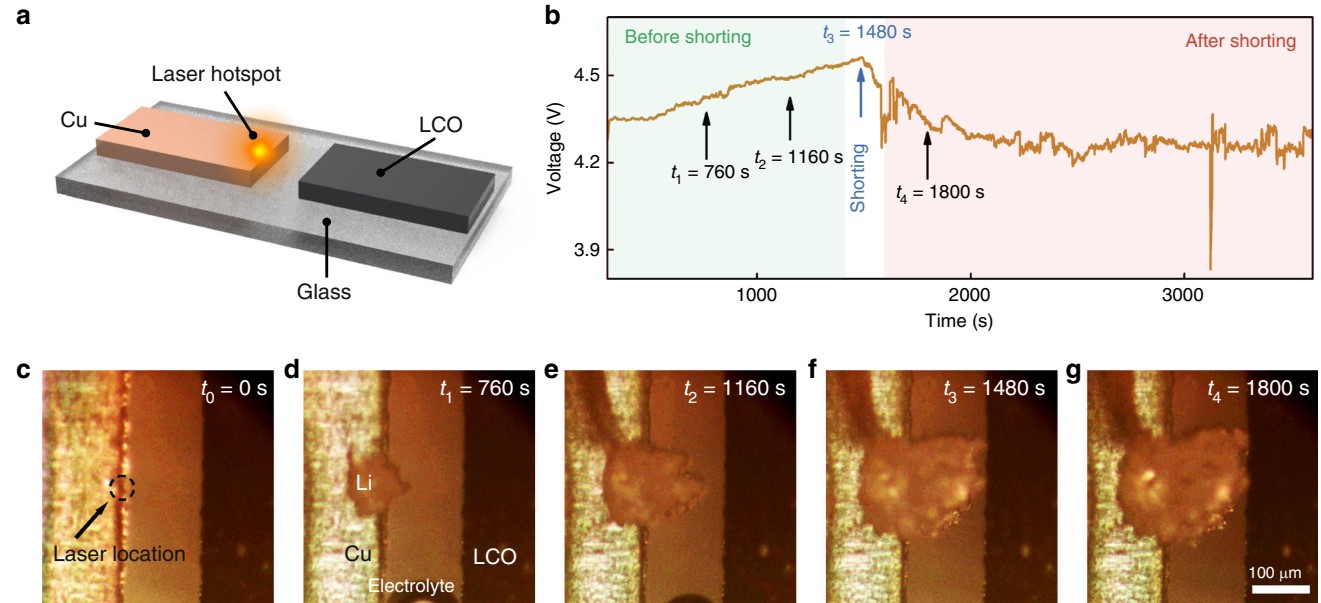

**Fig. 3** Hotspot-induced battery shorting. **a** Schematic of an optical cell with Cu and lithium cobalt oxide (LCO) as the electrodes. **b** Cell voltage as the battery was charged at a constant current of 30 μA. After onset of shorting, the voltage started to drop and fluctuate. Visualization of Li-plating process initially at **c** $t_0 = 0$ s, before shorting at **d** $t_1 = 760$ s, **e** $t_2 = 1160$ s, onset of shorting at **f** $t_3 = 1480$ s, and after shorting at **g** $t_4 = 1800$ s

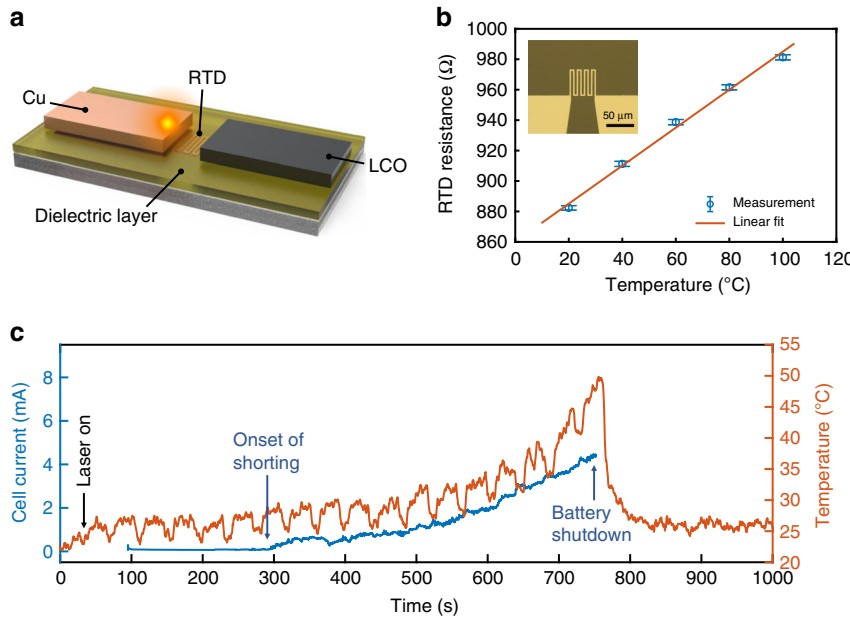

**Fig. 4** Hotspot-induced battery shorting and local-temperature response. **a** Schematic of the optical cell with a resistance temperature detector (RTD) in the copper-LCO (lithium cobalt oxide) gap and a laser hotspot. **b** Calibration of the RTD resistance as a function of the temperature. The dots are experimental measurements and the line is linear fit. **c** Current (left axis) of the battery and temperature response (right axis) measured by the RTD

beam on the Cu next to the RTD. The RTD was a platinum (Pt) thin-film (100 nm) pattern with a linear temperature-dependent electrical resistance, as shown in Fig. 4b. A polyimide (PI) film[33] (thickness of 1 μm) was spun and cured as a dielectric layer to insulate the RTD from the electrodes. The RTD offers faster temporal sampling rate than the Raman method, which was limited by the low Raman intensity of graphene, and is ideal for transient temperature sensing where spatial resolution of tens of microns (the size of the Li chunk) is acceptable. In addition, the slight misalignment of the heat source (on the Cu) and the temperature sensor (in the Cu–LCO gap) helps to partially decouple heating from the laser and heating as a result of battery

shorting. As shown in Fig. 4c, initially the battery was at room temperature ($t = 0$ s). The laser was turned on (13.4 mW) at $t = 30$ s before the battery started charging at $t = 95$ s at a constant voltage of 3.8 V. Thermal spreading of the hotspot caused the RTD temperature to rise only 5 °C. The periodic temperature dip was because the laser was turned off for 5 s every 40 s to allow optical visualization of the cell to monitor the charging process. Shorting occurred at ~300 s, followed by an increase in the current (Fig. 4c). Accordingly, temperature started to rise due to Joule heating from the local high current density. As more Li was accumulated at the shorting location, temperature increased to 50 °C. To prevent safety incidents and damage to the Raman

spectroscopy setup, charging was shut down at 50 °C, and the temperature reduced to room temperature. This method to measure the local-temperature response of a shorting event can serve as a tool to study other model battery systems. It should be noted that thermal runaway of a large size battery could be very different due to factors including cell capacity, architecture, and packaging, which affect heat spreading, among others. For real batteries, detection of early-stage internal thermal runaway or shorting may be possible by embedding RTD sensor networks into the battery, although the spatial resolution is limited by the spacings between adjacent RTDs.

## Discussion

To summarize, we investigated the effect of internal temperature hotspots on a lithium battery using micro-Raman spectroscopy as a temperature sensing platform. Li deposition rate was shown to be orders of magnitude faster on the hotspot due to the enhanced surface exchange current density. We further demonstrated with simultaneous voltage-current measurement, optical visualization, and temperature response that battery shorting can be triggered with a non-uniform, localized high temperature spot. The temperature-sensitive phenomena within lithium batteries highlighted in this work shed light on the positive feedback nature of Li dendrite growth; high local temperatures can initiate enhanced Li deposition rates, which can then further short the cell, raising local temperatures further. The two-way relationships between lithium dendrite growth and local temperature increases serve not only as foundations for understanding electrochemical dynamics within the cell, but also as guiding principles and limits for the design of practical cells. It should also be noted that in general, elevated cell temperatures can also trigger exothermic reactions of the electrolyte with active materials and the solid-electrolyte interphase (SEI)[7], further aggravating temperature increases. These findings suggest that the design of future high-power-density, fast-charging batteries need to take into consideration thermal management aspects to ensure uniform temperature, potentially via enhancing thermal conductivity of battery components, improving the tab design to reduce localization of Joule heating, minimizing defects, and utilizing efficient heat spreading in the current collector. In addition, temperature mapping techniques using micro-Raman spectroscopy or arrays of micro-RTDs can open new doors for detailed thermal characterization of energy storage devices. The insights gained from this study aid the understanding of battery failure mechanisms and the development of safer batteries, thermal management schemes, and diagnostic tools.

## Methods

**Coin cell**. For the laser-induced hotspot experiment, coin cells (CR 2032) were modified to have an optical window (1 cm$^2$ glass disk, thickness of 145 µm). Graphene was first exfoliated on a Si/SiO$_2$ substrate and then transferred onto the glass by polymer-assisted wet transfer. Subsequently, a 170 nm Cu layer was thermally evaporated on the glass, which covered the graphene and served as the working electrode. Li foil (thickness of 50 µm) was used as the counter electrode. Two layers of Celgard separators (Celgard 2325, 25 µm thick) were used to separate the working electrode and counter electrode. An additional polyimide ring disk (50 µm thick) was place between the working electrode and Celgard separator to prevent severe pressing pressure on the deposited lithium. 50 µL of 1 M LiPF$_6$ in 1:1 (v:v) ethylene carbonate (EC) and diethyl carbonate (DEC) (BASF) was added as the electrolyte. Battery testing was performed using a portable Biologic battery tester (SP-50 BioLogic). Raman spectroscopy was performed on a Horiba Labram HR Evolution Raman System. In the experiment, a constant amount of charge was applied at the same Li-plating rate of 1 mA/cm$^2$ for 2 min for coin cells with different laser heating power. After Li deposition, the coin cells were disassembled immediately inside the glove box. The optical windows deposited with Li were rinsed with diethyl carbonate to remove salt residues for SEM imaging (5 kV, FEI XL30 Sirion SEM).

**Optical cell**. In situ optical microscopy study was carried out using an optical cell with transparent windows. The working electrode (12 µm thick Cu foil) and the counter electrode (80% LiCoO$_2$, 10% Super P carbon, and 10% PVDF) were aligned

in parallel with a gap of ~100 µm. 1 M LiPF$_6$ in 1:1 (v:v) ethylene carbonate (EC) and diethyl carbonate (DEC) (BASF) was added as the electrolyte. The cell was covered with a 1 cm$^2$ cover glass (thickness of 145 µm) and the edges were sealed with epoxy. The fabricated optical cells were characterized on the Raman spectroscopy platform. During lithium deposition, an optical image (through a ×10 objective) of the Li deposited on the Cu working electrode was recorded every 40 s, and for the rest of time the light source was switched to a 532 nm laser, which produced a hotspot on the Cu.

**Resistance temperature detectors**. Resistance Temperature Detectors (RTDs) were fabricated on glass slide substrates (Corning, 2947–75 × 50). The RTD pattern was defined through a standard photolithography process. Subsequently, a 15 nm titanium (Ti) and a 100 nm platinum (Pt) were thermally evaporated onto the substrates and the photoresist was removed by overnight soak in acetone. The Pt RTDs were annealed at 300 °C for 1 h to avoid resistance drift. A 1 µm thick polyimide layer was spin coated and cured to cover the RTD as an electrical insulation layer (detailed synthesis procedure for the polyimide layer can be found in ref. [33]). All the RTDs were calibrated in an environmental chamber (BTU-133, ESPEC). The resistance of the RTDs was measured at least 1 hour after each set temperature to allow adequate time for the samples to reach thermal equilibrium.

## Data availability

The data that support the findings of this study are available from the corresponding author upon request.

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

## Acknowledgements

This work was supported by the Assistant Secretary for Energy Efficiency and Renewable Energy, Office of Vehicle Technologies of the U.S. Department of Energy under the eXtreme Fast Charge Cell Evaluation of Li-ion batteries (XCEL) program. Jin Xie acknowledges support from ShanghaiTech University start-up funding. The authors would like to thank Dr. Andrey Malkovskiy at Stanford and Lenan Zhang at MIT for valuable discussion on Raman spectroscopy. Part of this work was performed at the Stanford Nano Shared Facilities and the Stanford Nanofabrication Facility.

## Author contributions

Y.Z. and J.X. contributed equally to this work. Y.Z., J.X., and Y.C. conceived the initial idea. Y.Z. and J.X. designed and conducted the experiment. A.P. and Y.Z. performed COMSOL simulation. B.L., Y.W., D.L., and J.L. helped with fabricating the resistance temperature detectors, graphene, and polyimide. Hs.W., H.C., Jw.X., W.C., and Hx.W. provided discussion on the experiment. A.Y. and C.W. performed thin-film metal deposition. Y.Z., J.X, A.P., and Y.C. wrote the manuscript and all authors discussed it before submission.

## Additional information

**Competing interests:** The authors declare no competing interests.

