## [Peer Review File · Nature Communications]

REVIEWERS' COMMENTS:

Reviewer #1 (Remarks to the Author):

The authors have made extensive revisions in response to my previous comments. In several cases they have provided even more information than requested, via additional simulations in the SI, and I applaud them for this thoroughness and attention to detail.

I am largely in agreement with the revisions and the arguments provided in the response letter, and now recommend this work for publication in Nature Communications.

Reviewer #2 (Remarks to the Author):

The design of the experimental platforms in this research is very novel and the localized hotspot has been found to cause internal shorting of battery in this study. In the revised manuscript, the authors have answered the reviewer's comments in detail with reasonable explanations and the manuscript has been revised accordingly. I think it is suitable for the publication in Nature Communications with revision required. The manuscript is presented in a clear way but some of claims seem are not well-illustrated.

1. For the novelty, the design of the experiment is innovative. These findings suggest that the design of future high-power-density, fast-charging batteries need to take into consideration thermal management aspects to ensure uniform temperature. This seems not a conclusion but more as a common-sense-like recommendation. Which properties can be contributed to practical applications.
2. The authors convinced that these conclusions can be extended to cylindrical batteries pouch cells, cylindrical cells and prismatic cells. Although obtaining the exact distribution for the locally enhanced Li growth rate for different systems was not the focus of this study. The thermal runaway of coin cells and large size batteries should be very different.
3. "To ensure that shorting occurs precisely on top of the RTD, we aligned the laser beam on the Cu next to the RTD." "Spatial detection of early-stage internal thermal runaway or shorting in large size batteries may be possible by embedding RTD arrays into the battery" The position of the short circuit inside the battery is difficult to accurately determine, in this case, how to ensure the installation location of the RTD in practical applications.

Author response to reviewers' comments

Manuscript ID: NCOMMS-19-05939-T

Title: Fast lithium growth and short circuit induced by localized-temperature hotspots in lithium batteries

Authors: Yangying Zhu, Jin Xie, Allen Pei, Bofei Liu, Yecun Wu, Dingchang Lin, Jun Li, Hansen Wang, Hao Chen, Jinwei Xu, Ankun Yang, Chun-Lan Wu, Hongxia Wang, Wei Chen, Yi Cui

The authors are grateful to the reviewers for their constructive comments on the resubmitted manuscript. All reviewers' comments were carefully considered and implemented/responded in the further revised manuscript. A detailed response is given for each comment. Reviewers' comments are shown in blue italics.

Reviewer #1 (Remarks to the Author):

The authors have made extensive revisions in response to my previous comments. In several cases they have provided even more information than requested, via additional simulations in the SI, and I applaud them for this thoroughness and attention to detail.

I am largely in agreement with the revisions and the arguments provided in the response letter, and now recommend this work for publication in Nature Communications.

Response: We appreciate the reviewer's detailed comments which helped to significantly improve the quality of this paper.

Reviewer #2 (Remarks to the Author):

The design of the experimental platforms in this research is very novel and the localized hotspot has been found to cause internal shorting of battery in this study. In the revised manuscript, the authors have answered the reviewer's comments in detail with reasonable explanations and the manuscript has been revised accordingly. I think it is suitable for the publication in Nature Communications with revision required. The manuscript is presented in a clear way but some of claims seem are not well-illustrated.

Response: We appreciate the reviewer's positive evaluation of this work, and suggestions to improve the accuracy of a few claims.

1. For the novelty, the design of the experiment is innovative. These findings suggest that the design of future high-power-density, fast-charging batteries need to take into consideration thermal management aspects to ensure uniform temperature. This seems not a conclusion but more as a common-sense-like recommendation. Which properties can be contributed to practical applications.

Response: We appreciate and agree with the reviewer's suggestion. We have revised the statement to be more specific: localized temperature hotspots and temperature heterogeneity should be

minimized in the design of future batteries. Possible means to mitigate temperature heterogeneity are also added (Page 12):

“These findings suggest that the design of future high-power-density, fast-charging batteries need to take into consideration thermal management aspects to ensure uniform temperature, potentially via enhancing thermal conductivity of battery components, improving the tab design to reduce localization of Joule heating, minimizing defects, and utilizing efficient heat spreading in the current collector.”

2. The authors convinced that these conclusions can be extended to cylindrical batteries pouch cells, cylindrical cells and prismatic cells. Although obtaining the exact distribution for the locally enhanced Li growth rate for different systems was not the focus of this study. The thermal runaway of coin cells and large size batteries should be very different.

Response: For the conclusions that can be extended to other battery forms (cylindrical, prismatic and pouch cells), we refer solely to the finding that temperature hotspots enhance lithium deposition which could trigger shorting. We agree with the reviewer that thermal runaway of coin cells and large size batteries are different due to factors including cell capacity, architecture and packaging which affect heat spreading, among others, and therefore, we did not discuss thermal runaway temperature response of other cells in the manuscript. To clarify, we have added (Page 11):

“It should be noted that thermal runaway of a large size battery could be very different due to factors including cell capacity, architecture, and packaging which affect heat spreading, among others.”

3. “To ensure that shorting occurs precisely on top of the RTD, we aligned the laser beam on the Cu next to the RTD.” “Spatial detection of early-stage internal thermal runaway or shorting in large size batteries may be possible by embedding RTD arrays into the battery” The position of the short circuit inside the battery is difficult to accurately determine, in this case, how to ensure the installation location of the RTD in practical applications.

Response: In the manuscript we provide a method to thermally trigger shorting exactly where we place a temperature sensor for an optical cell. We agree that the position of short circuit in real batteries is difficult to determine. The RTD arrays inside the battery could sense thermal runaway after heat from the shorting location has spread to the nearest sensor. The more RTDs embedded inside the battery the quicker it is able to sense an internal thermal runaway due to a shorter heat spreading distance. We have modified the statement as (Page 11):

“For real batteries, detection of early-stage internal thermal runaway or shorting may be possible by embedding RTD sensor networks into the battery, although the spatial resolution is limited by the spacings between adjacent RTDs.”